

# High resolution biogenic global emission inventory for the time period 2000-2019 for air quality modelling

Katerina Sindelarova[1], Jana Markova[1,2], David Simpson[3,4], Peter Huszar[1], Jan Karlicky[1], Sabine Darras[5], Claire Granier[6,7]

[1]Charles University, Faculty of Mathematics and Physics, Dept. of Atmospheric Physics, Prague, Czechia
[2]Czech Hydrometeorological Institute, Prague, Czechia
[3]Climate Modelling and Air Pollution Division, Research and Development Dept. Norwegian Meteorological Institute, Oslo, Norway
[4]Dept. Space, Earth & Environment, Chalmers Univ. Tech. Gothenburg, Sweden
[5]Observatoire Midi-Pyrénées, Toulouse, France
[6]Laboratoire d'Aérologie, Université de Toulouse, CNRS, UPS, Toulouse, France
[7]NOAA Chemical Sciences Laboratory and CIRES, University of Colorado Boulder, Boulder, Colorado, USA

*Correspondence to*: Katerina Sindelarova (katerina.sindelarova@mff.cuni.cz)

**Abstract.** Biogenic volatile organic compounds (BVOCs) emitted from the terrestrial vegetation into the Earth's atmosphere play an important role in atmospheric chemical processes. A gridded information of their temporal and spatial distribution is therefore needed for proper representation of the atmospheric composition by the air quality models. Here we present three newly developed high-resolution global emission inventories of the main BVOC species including isoprene, monoterpenes, sesquiterpenes, methanol, acetone and ethene. Monthly mean and monthly averaged daily profile emissions were calculated by the Model of Emission of Gases and Aerosols from Nature (MEGANv2.1) driven by meteorological reanalyzes of the European Centre for Medium-Range Weather Forecasts for the period of 2000-2019. The dataset CAMS-GLOB-BIOv1.2 is based on ERA-Interim meteorology, datasets CAMS-GLOB-BIOv3.0 and v3.1 were calculated with ERA5. Furthermore, European isoprene emission potential data were updated using high-resolution land cover maps and detailed information of tree species composition and emission factors from the EMEP MSC-W model system. Updated isoprene emissions are included in CAMS-GLOB-BIOv3.1 dataset. The effect of annually changing land cover on BVOC emissions is captured by the CAMS-GLOB-BIOv3.0 as it was calculated with land cover data provided by the Climate Change Initiative of the European Space Agency (ESA-CCI). The global total annual BVOC emissions averaged over the simulated period vary between the datasets from 424 to 591 Tg(C) yr$^{-1}$, with isoprene emissions from 299.1 to 440.5 Tg(isoprene) yr$^{-1}$. Differences between the datasets and variation in their emission estimates suggests the emission uncertainty range and the main sources of uncertainty, i.e. meteorological inputs, emission potential data and land cover description. The CAMS-GLOB-BIO time series of isoprene and monoterpenes were compared to other available data. There is a general agreement in an inter-annual variability of the emission estimates and the values fall within the uncertainty range. The CAMS-GLOB-BIO datasets (CAMS-GLOB-BIOv1.2, https://doi.org/10.24380/t53a-qw03, Sindelarova et al., 2021a; CAMS-GLOB-BIOv3.0, https://doi.org/10.24380/xs64-gj42, Sindelarova et al., 2021b; CAMS-GLOB-BIOv3.1, https://doi.org/10.24380/cv4p-5f79, Sindelarova et al., 2021c) are



distributed from the Emissions of atmospheric Compounds and Compilation of Ancillary Data (ECCAD) system
(https://eccad.aeris-data.fr/, last access: June 2021).

## 1 Introduction

The biogenic organic volatile compounds (BVOCs) consist of a vast group of non-methane hydrocarbons emitted from
terrestrial vegetation and soils into the Earth's atmosphere (Kesselmeier and Staudt, 1999). They form about 90% of the total
atmospheric VOC budget (Guenther et al., 1995) and due to their high reactivity, they are an important component of
atmospheric chemistry (Atkinson and Arey, 2003). Through their oxidation in the atmosphere, BVOCs affect tropospheric
photochemistry and composition (Houweling et al., 1998; Williams et al., 2013).

Their oxidation products play an important role in formation of low-level ozone and secondary organic aerosols thus having
impact on air quality and Earth's radiative budget. Their impact on tropospheric ozone levels was evaluated by series of
modelling studies on both global (e.g. Poisson et al., 2000; Pfister et al., 2008) and regional scales (e.g. Curci et al., 2009;
Sartelet et al., 2012; Situ et al., 2013; Tagaris et al., 2014). The evidence of formation of secondary organic aerosols from
BVOC oxidation products was observed by experimental studies (e.g. Griffin et al., 1999; Hao et al., 2011) and field studies
(Lemire et al., 2002; Gelencser et al., 2007; Ehn et al., 2014), and consequently evaluated by atmospheric chemistry models
(e.g. van Donkelaar et al., 2007; Simpson et al., 2007; Hodzic et al., 2010; Wu et al., 2020).

The emission of BVOCs from vegetation depends on many environmental factors such as meteorology, especially air
temperature and solar radiation, type of vegetation, seasonal cycle, atmospheric chemistry and composition (Guenther et al.,
1995) and therefore varies significantly in space and time. As biosphere-atmosphere interaction is a very complex system with
mutual feedbacks, efforts have been made to assess the impact of different driving factors which themselves are changing
and/or are expected to change in the future. An interest has been focused mainly on impact of changing climate, land cover
and atmospheric $CO_2$ concentration in the recent past (e.g. Naik et al., 2004; Lathière et al., 2006; Arneth et al., 2007a;
Stavrakou et al., 2014) as well as in the future (e.g. Sanderson et al., 2003; Heald et al., 2008; Hantson et al., 2017).

Given the importance of BVOCs in the atmospheric composition, a proper information of amount and spatio-temporal
distribution of BVOC emissions is a crucial input to atmospheric chemistry and climate models. Different ground-based
measurement techniques can be applied to sample BVOC emissions at different scales, from leaf to regional level, as
summarized by Hewitt et al. (2011). Many measurement campaigns have been organized in the past to evaluate BVOC
emission fluxes at different parts of the world, especially in the tropics which have the highest emission potential such as the
Amazon (e.g. Rinne et al., 2002; Kuhn et al., 2007; Karl et al., 2007; Eerdekens et al., 2009) and Southeast Asia (e.g. Langford
et al., 2010, Misztal et al., 2011). However, such measurements are unfortunately limited in space and time and are therefore

not fully suitable to create a long-term gridded inventory of BVOC emissions required by the models. Knowledge obtained from observations on the emission processes, speciation and evaluation of fluxes serves as a valuable baseline for development of the emission BVOC models which are then able to simulate BVOC emissions for a specific time period and spatial domain

based on defined input parameters.

Over the time a relatively long list of BVOC emission models have been developed. The models differ in used approach to BVOC estimation and level of complexity in considered processes and factors affecting the emission. In general, there are two main approaches to BVOC modelling. First, a so-called *process-based model* that simulates BVOC synthesis directly inside

the plant (e.g. LPJ-GUESS, JULES). Second, based on a semi-empirical algorithm described by Guenther et al. (1995) which defines dependence of BVOC emissions from the plant on environmental factors, namely air temperature and solar radiation. From the latter developed the MEGAN model, widely used in the BVOC emission and atmospheric chemical and climate modelling communities. The emission algorithms can either be used as stand-alone, or can be embedded inside an Earth system, land surface or air quality model.


Different BVOC emission models were applied in the past to obtain estimates of BVOC emission levels on global scale (e.g. Lathière et al., 2005; Müller et al., 2008; Arneth et al., 2007b; Schurgers et al., 2009; Pacifico et al., 2011; Guenther et al., 2012; Sindelarova et al., 2014; Messina et al., 2016). Similarly, there exists long list of studies focusing on regional level (e.g. Simpson et al., 1995; Simpson et al., 1999; Steinbrecher et al., 2009; Karl et al., 2009; Oderbolz et al., 2013; Emmerson et al.,

2018). These inventories are so called 'bottom-up', i.e. calculated by the emission models based on surface input data.

With emerging availability of satellite-based observations of the Earth's atmosphere, data retrieved from space started to be used also in BVOC emission estimation. Space-borne measurements of suitable chemical species are used to constrain a-priori emissions through an inversion technique applied in the atmospheric chemistry model. Such approach has been applied for

example to constrain emissions of isoprene, the most abundant BVOC species, with satellite measurements of isoprene's oxidation product formaldehyde (e.g. Palmer et al., 2006; Millet et al., 2008; Stavrakou et al., 2009; Curci et al., 2010; Bauwens et al., 2016; Kaiser et al., 2018). Emission inventories constrained by satellite observations through application of the model inversion are being called 'top-down'. Recently, a methodology for direct measurement of isoprene emissions from space has been developed by identifying spectral signatures of isoprene in satellite-borne measurements of Cross-track Infrared Sounder

(Fu et al., 2019; Wells et al., 2020).

In this paper we present three new global bottom-up inventories of BVOC emissions calculated with modified version of the state-of-the-art emission model MEGANv2.1 (Guenther et al., 2012) forced by meteorological reanalyzes of the European Centre for Medium-Range Weather Forecasts (ECMWF) and high-resolution input data. The inventories contain global





gridded emissions of the main BVOC species for a 20-year period (2000-2019) which can be directly used as an input to the air quality and climate models.

In the following section (Sect. 2) we describe a methodology of emission calculation, including description of the emission model, input meteorological, land cover and emission factor data. Sect. 3 presents global and regional distribution of emission estimates, together with comparison of emission inventories within each other and with other available data. Information on data availability is given in Sect. 4 and conclusions and summary are presented in Sect. 5.

## 2 Methodology

### 2.1 Emission model

The presented emission datasets were calculated using the Model of Emissions of Gases and Aerosols from Nature (MEGANv2.1, Guenther et al., 2012). The MEGAN model was developed at the National Center for Atmospheric Research (NCAR, US) and is currently maintained and further improved by Biosphere Atmosphere Interaction Group at University of California – Irvine (https://bai.ess.uci.edu/).

It is an emission model extensively used in the atmospheric modelling community for simulation of biogenic VOC emissions from vegetation and soils at regional and global scales (e.g. Guenther et al., 2006; Heald et al., 2008; Arneth et al., 2011; Sindelarova et al., 2014; Seco et al., 2015; Emmerson et al., 2018; Kaiser et al., 2018, Huszar et al, 2018, 2020). Furthermore, the algorithm of the MEGAN model has been embedded into number of Earth system and chemical transport models (e.g. Emmons et al., 2010; Lawrence et al., 2011; Keller et al., 2014; Henrot et al, 2017).

The model calculates an emission flux $F$ ($\mu$g grid cell$^{-1}$ h$^{-1}$) of specific BVOC species from a model grid cell as follows:

$$F = \gamma \, . \, EP \, . \, S \tag{1}$$

where $\gamma$ is a dimension-less factor accounting for dependence of emissions on environmental factors (air temperature, solar radiation, ambient $CO_2$ concentration, leaf age, etc.), $EP$ ($\mu$g m$^{-2}$ h$^{-1}$) is an emission potential of a grid cell, i.e. a unit emission defined under standardized environmental conditions and $S$ (m$^2$) is a grid cell surface area. The MEGANv2.1 was applied with the full canopy module which calculates meteorological conditions inside the forest canopy (e.g. leaf temperature, radiation on sunlit and shaded leaves). For calculation of isoprene, the model took into account an inhibitory effect of $CO_2$ concentration on isoprene emissions using parametrization described in Heald et al. (2009). In our simulations, we did not consider the effect of soil moisture stress on the plant emissions. For more details on the MEGANv2.1 algorithm please see Guenther et al. (2006, 2012).



## 2.2 Meteorology

Two sources of meteorological data were used for calculation of the emission datasets. CAMS-GLOB-BIOv1.2 is based on the ERA-Interim (Dee at al., 2011) data and datasets CAMS-GLOB-BIOv3.0 and v3.1 were calculated with ERA5 (Hersbach et al., 2020), both meteorological reanalyzes of the European Centre for Medium-Range Weather Forecasts (ECWMF).

MEGAN model requires the following input parameters – 2 m air temperature, water mixing ratio, surface pressure, 10 m wind speed and photosynthetically active radiation (PAR). PAR is defined as solar radiation with wavelength between 400 and 700 nm which photosynthetic organisms are able to absorb during photosynthesis. Unfortunately, this parameter is not available in ERA-Interim nor ERA5 datasets (see Copernicus Knowledge Base – ERA-Interim: surface photosynthetically active radiation (surface PAR) values are too low, 2017). PAR was therefore approximated with surface solar downward radiation divided by

a factor of 2.2 as recommended by various studies (Olofsson et al., 2007; Jacovides et al., 2003; Escobedo et al., 2011). The water mixing ratio was calculated from 2 m dew point temperature following equations from Lowe and Ficke (1974).

Since emissions are calculated on a monthly mean basis, the input meteorological data were synoptic monthly means of analyzed and forecasted parameters. ERA-Interim data were available on global grid with horizontal spatial resolution of $0.5°$

x 0. 5° with 3 or 6 h time steps. The data were linearly interpolated in time in order to obtain monthly averaged daily profile of each meteorological variable. ERA5 is a successor to ERA-Interim with higher horizontal spatial resolution of $0.25° \times 0.25°$ and with 1 h time resolution. Interpolation between time steps was therefore no longer necessary in case of ERA5.

## 2.3 Vegetation description

The spatial distribution of vegetation in the MEGAN model is defined using plant functional types (PFTs). This is an alternative

approach to vegetation description using biomes (e.g. savanna, tundra). While biomes can consist of physiologically distinct vegetation types (e.g. grasses and trees), plant functional types group vegetation with similar leaf physiology. Use of PFTs leads to less complex vegetation representation, but allows physiologically-based ecosystem description convenient for the dynamic global vegetation models. The MEGAN model was designed to be coupled with Community Land model (CLM4) and therefore uses the same approach, i.e. representation of the global land cover with 16 PFT categories (Lawrence and Chase,

2007). Vegetation in each model grid cell is defined by fractional coverage by each of the PFT. List of the MEGANv2.1 PFT categories is given in Table 1.

Emissions in CAMS-GLOB-BIOv1.2 and v3.1 were calculated with temporally invariable map of PFTs from CLM4 model representative for the year 2000. However, global land use / land cover is experiencing dramatic changes, e.g. deforestation in

the tropical forests and replacement of forests by agricultural land (e.g. Song et al., 2018), which is obviously expected to impact the BVOC emissions. In order to capture the land cover change in MEGAN simulations, we replaced the static CLM4 PFT map with land cover data from the ESA-CCI (ESA, 2017). ESA-CCI data are provided by the Climate Change Initiative



of the European Space Agency. The data consists of time series of global annual mean land cover maps with high horizontal spatial resolution (300 m) available for the period of 1992-2018 based on satellite observations. To be consistent with the

MEGAN model, the ESA-CCI land cover categories were converted to PFT classes similar to CLM using the CCI-LC user tool v4.3 (Poulter et al., 2015). Emissions calculated with temporally varying land cover are included in CAMS-GLOB-BIOv3.0 dataset.

Table 1 compares global land areas covered by each PFT category in CLM4 and ESA-CCI (year 2000, converted by CCI-LC

user tool) land cover maps. Note that though Corn (Maize) category is included in the MEGAN PFT list, it is currently not distinguished from other crops and its spatial coverage is therefore zero. The two maps differ in the total area covered by vegetation, with ESA-CCI giving ~19% less vegetated area globally. In ESA-CCI, extent of a tree and grass categories is ~25% lower, while coverage by the crop category is almost 50% higher when compared to CLM.

Vegetation seasonality is represented by changes in leaf area index (LAI). LAI is a dimensionless parameter defined as one-sided leaf area per area of the ground surface ($m^2$ $m^{-2}$). Spatial and temporal distribution of LAI was obtained from processed observations of the MODIS instrument (Yuan et al., 2011). The 8-day observations were averaged to monthly means. The Yuan et al. LAI data are available for the period of 2000 – 2016. For the emissions calculated after 2016 a 10-year climatology was used. Since these LAI values represent an average over the whole grid cell, values were divided by a grid cell fraction

covered by vegetation to obtain LAI for vegetated area only.

**Table 1. List of MEGANv2.1 PFT categories and global coverage by each PFT category ($10^6$ $km^2$) according to CLM4 and ESA-CCI (year 2000) land cover maps.**

| PFT category | global area / $10^6$ $km^2$ | |
|---|---|---|
| | CLM4 | ESA-CCI 2000 |
| Needleleaf evergreen temperate tree | 3.63 | 3.03 |
| Needleleaf deciduous boreal tree | 1.48 | 3.05 |
| Needleleaf evergreen boreal tree | 9.92 | 3.51 |
| Broadleaf evergreen tropical tree | 11.83 | 10.25 |
| Broadleaf evergreen temperate tree | 1.91 | 2.42 |
| Broadleaf deciduous tropical tree | 6.13 | 3.33 |
| Broadleaf deciduous temperate tree | 4.63 | 3.64 |
| Broadleaf deciduous boreal tree | 1.76 | 1.24 |





| | | |
|---|---|---|
| Broadleaf evergreen temperate shrub | 0.09 | 1.73 |
| Broadleaf deciduous temperate shrub | 5.49 | 3.09 |
| Broadleaf deciduous boreal shrub | 8.18 | 1.52 |
| Arctic C3 grass | 4.31 | 5.96 |
| Cool C3 grass | 12.67 | 7.97 |
| Warm C4 grass | 11.20 | 6.79 |
| Crops | 14.76 | 21.90 |
| Corn (Maize) | 0 | 0 |
| *Total* | *98.0* | *79.4* |


## 2.4 Global emission potential data

Emission potentials are together with the vegetation description a crucial parameter in BVOC emission estimation. In the following text we distinguish between emission factor (EF) and emission potential (EP). By EF we mean emission of a

chemical species from specific plant or vegetation type under standard conditions of environmental parameters. EFs can be defined either as area-based values, i.e. an emission from a unit area covered by specific plant or vegetation type (e.g. $\mu g$(species) $m^{-2}$ (ground-cover) $h^{-1}$) or mass-based values, i.e. emission from a unit mass of the plant's dry leaf matter (e.g. $\mu g$(species) $g^{-1}$ (dry foliar mass) $h^{-1}$). With emission potential (EP, $\mu g\ m^{-2}\ h^{-1}$) we describe the emission capacity of the whole grid cell which is calculated as a weighted sum of emission factors for all plants or vegetation types present in the grid cell:


$$EP = \sum_{veg} f_i EF_i (\mu g\ m^{-2} h^{-1}) = \sum_{veg} f_i EF_i (\mu g\ g^{-1}\ h^{-1})\, D_i \tag{2}$$

where $f_i$ is a fraction of a grid cell covered by individual plant or vegetation type and $D_i$ (g(dry leaf matter) $m^{-2}$) is a foliar density of the plant or vegetation type.


Emission factors in the MEGANv2.1 model are defined on a canopy-scale level as an emission under standard condition from the full canopy. Above canopy measurements of EF are unfortunately limited, therefore the canopy-scale EFs in MEGAN are still based on leaf- and branch-scale measurements which were extrapolated with a canopy environment model to the canopy level (Guenther et al., 2006). MEGAN standard conditions are defined for series of variables, such as LAI, leaf age composition

of the canopy, meteorological conditions (temperature, solar radiation, humidity, wind speed, soil moisture) of the current state and of the past (temperature and solar radiation). For more details see Guenther et al. (2006).

The MEGAN model has two options for emission potential definition. Either use of the input emission potential maps for selected species or calculation of EP from vegetation coverage. These options are described in more detail in the two following sections.

### 2.4.1 Emission potentials from detailed global maps

The first option consists of the use of annual mean emission potential maps with high spatial resolution for the main BVOC species, i.e. isoprene, main monoterpenes (α-pinene, β-pinene, myrcene, sabinene, limonene, trans-β-ocimene, 3Δ-carene) and 2-methyl-3-buten-2-ol (MBO). Emission maps are available together with the MEGANv2.1 code (https://bai.ess.uci.edu/megan/data-and-code/megan21, access date 31/5/2021) and were created based on detailed ecoregion description, combining information on species composition with species specific emission factors and above canopy flux measurements, where available (Guenther et al., 2012). Emission potentials for the rest of the modeled species were calculated based on the PFT coverage as described in the following section.

### 2.4.2 Emission potentials calculated from PFTs

The second option consists of EP calculation from the vegetation composition of each grid cell. MEGAN uses 16 PFTs for description of vegetation in the model domain (listed in Table 1.). Each of the PFTs is assigned with an emission factor value for each of the modeled species (see Table 2 in Guenther et al., 2012). The emission potential of each grid cell for a specific modeled chemical species is then calculated as a weighted sum defined in Eq. (2).

Use of EP calculated from the PFT coverage leads to ~10% decrease of isoprene emission total on global scale when compared to emission calculation based on EP detailed maps. For β-pinene and other monoterpenes the difference is only 1-2 %. However, for α-pinene emissions calculated from PFT coverage are more than 70 % higher when compared to emissions calculated from the EP maps. Therefore, the EFs assigned to each PFT tree category for α-pinene were revised based on recent updates of EFs for the ORCHIDEE model (Messina et al., 2016). The ORCHIDEE and MEGAN models differ in definition of standard conditions which means that the ORCHIDEE EFs needed to be converted to MEGAN suitable format. The conversion was done in a similar way as described in Sect. 2.5.2. The newly used α-pinene EFs are listed in Table 2 together with the original MEGANv2.1 values. The resulting α-pinene emissions calculated with revised EFs are ~ 18% higher than emissions calculated with detailed EP map.





**Table 2. Updated emission factors for α-pinene (μg m$^{-2}$ h$^{-1}$) for tree PFT classes used in this study based on Messina et al. (2016) together with original MEGANv2.1 EFs (Guenther et al., 2012).**

| | | EF α-pinene (μg m$^{-2}$ h$^{-1}$) | |
|---|---|---|---|
| PFTS | Description | this study / Messina et al. (2016) | Guenther et al. (2012) |
| NT_EG_TEMP | Evergreen needleleaf temperate | 373 | 500 |
| NT_DC_BORL | Deciduous needleleaf boreal | 698 | 510 |
| NT_EG_BORL | Evergreen needleleaf boreal | 373 | 500 |
| BT_EG_TROP | Evergreen broadleaf tropical | 386 | 600 |
| BT_EG_TEMP | Evergreen broadleaf temperate | 380 | 400 |
| BT_DC_TROP | Deciduous broadleaf tropical | 386 | 600 |
| BT_DC_TEMP | Deciduous broadleaf temperate | 204 | 400 |
| BT_DC_BORL | Deciduous broadleaf boreal | 259 | 400 |


Describing global vegetation by only 16 PFT categories is of course a simplification that inevitably brings inaccuracies, especially for categories such as broadleaf deciduous forest which can consist of tree species which are very low isoprene emitters but at the same time tree species such as oaks which are very strong isoprene emitters. On the other hand, such simplifications are often necessary due to lack of detailed information on vegetation composition and/or assignment with

emission factor, or is simply a result of balance between the level of detail in vegetation description and ability of the model algorithm to digest such data.

Calculation of EP from PFT coverage is to some extent inaccurate, on the other hand allows us to change the land cover description dataset (e.g. use ESA-CCI instead of CLM) and therefore study impact of land cover on resulting emissions.

**2.5 Update of isoprene emission potentials in Europe**

The MEGAN global input emission potential maps for isoprene and main monoterpenes were created based on information of global land cover distribution and vegetation composition in combination with emission factor survey, incorporating results of flux measurement campaigns. Naturally, lot of information on emission factor and flux measurements originates in the tropics as it is a region of the highest emission rates. This leads to the fact that the MEGAN emission potential maps are well suited

to for the tropical region, but may be less fitting in other parts of the world.

As detailed land cover data were becoming available for Europe, studies focusing on estimation of biogenic VOCs from plant-specific vegetation description started to appear (e.g. Simpson et al., 1995, 1999, Karl et al., 2009; Oderbolz et al., 2013).



Several studies have shown large discrepancies between emissions calculated using species-specific emission factors and those
calculated by MEGAN-based inputs (Rinne et al., 2009; Langner et al., 2012; Jiang et al., 2019). This motivated us to revise
the input emission potential maps for isoprene in this region.

In this work, new maps of area-based isoprene emission potentials (EP, $\mu g\ m^{-2}\ h^{-1}$) for the European area were created. These
EP maps are based on detailed maps of forest species and other vegetation combined with Europe-specific emission factors
for each species. The EP map update makes use of procedures developed over many years for the EMEP model (Simpson et
al., 1995, 1999, 2012). The basic emission factors and LAI changes are taken from a high-resolution version of the EMEP
model rv4.33 (Simpson et al., 2012).

The EMEP and MEGANv2.1 models differ in their definition of standard environmental conditions for emission factors. In
MEGANv2.1 EFs are defined on canopy-scale level, i.e. as an emission from the full canopy, under standardized canopy
conditions of LAI, specific proportion of mature, growing and old foliage, current and previous air temperatures and radiation,
humidity, wind-speed and soil moisture (Guenther et al., 2006, 2012). The EMEP system, similar to previous BVOC
algorithms of Guenther et al. (1995), uses leaf- and branch-level EF definition with standard conditions for leaf temperature
(30°C) and photosynthetically active radiation (1000 $\mu mol\ m^{-2}\ s^{-1}$) only. As canopy-scale EFs are not available for the
vegetation species used for this isoprene EP update, a new map was created with leaf- and branch-level EFs. The new isoprene
EP values were then converted to MEGANv2.1-suitable format. There is unfortunately no accurate conversion equation that
would satisfy all conditions. A rough conversion can be made following recommendations by Arneth et al. (2011) and Messina
et al. (2016) for conversion between the two systems. Details of the conversion of EMEP EFs to MEGANv2.1-suitable isoprene
EPs are given in Sect. 2.5.2.

### 2.5.1 Land cover description and emission factors

The main basis for European BVOC emissions in Europe in the EMEP system is a map of forest species generated by Köble
and Seufert (2001), combined with species-specific EFS for each of these species. The forest database provides maps for 115
tree species in 30 (mainly EU) European countries, based on a compilation of data from the ICP-forest network UN-ECE
(1998). These data were further processed to the EMEP grid by the Stockholm Environment Institute at York (UK, S. Cinderby
pers. Comm., 2004) in order to add data from other countries in the (2000-era) EMEP domain, and for non-forested vegetation.
More recently, the EMEP domain was significantly expanded to the east, and data for the expanded area and indeed globally
makes use of a merge of the GLC_2000 dataset (http://bioval.jrc.ec.europa.eu/products/glc2000/glc2000.php) and data from
the Community Land Model (http://www.cgd.ucar.edu/models/clm/, Oleson et al. 2010, Lawrence et al. 2011) as described in
Simpson et al. (2017). In order to provide a manageable number of PFTs for use in MEGAN, tree species were aggregated in
six classes, as summarized in Table 3.



**Table 3. Generic PFTs used for European emission potential maps based on EMEP.**

| PFT | Vegetation included | Examples | LAI variation | LAI max / $m^2\, m^{-2}$ |
|-----|---------------------|----------|---------------|--------------------------|
| CF | Temperate/boreal coniferous forest | norway spruce, scots pine | constant | 5 |
| DF | Temperate/boreal deciduous forest | european oak, beech, birch | variable | 4 |
| NF | Mediterranean needleleaf forest | cedars, eucalyptus, stone pine | constant | 4 |
| BF | Mediterranean broadleaf forest | holm oak, cork oak, arbutus | constant | 4 |
| SNL | Seminatural | moorland, tundra, shrub | variable | 3 |
| CR | Crops | all crops | variable | 3.5 |

For each grid cell, the grid-average emission potential of a specific PFT ($EP_{PFT}$, µg $m^{-2}$ $h^{-1}$) was then calculated as a weighted average of all the individual tree species belonging to this PFT category:

$$EP_{PFT} = \frac{\sum_i EF_i D_i A_i}{\sum_i A_i} \qquad (3)$$

where $i$ represents one of the many forest or vegetation species contained within that PFT, $EF_i$ is the species foliage-level isoprene emission factor (µg g(dry leaf weight)$^{-1}$ $h^{-1}$), $D_i$ is the species foliar density (g(dry leaf weight) $m^{-2}$) and $A_i$ is the species area ($m^2$). Further details of this methodology, including detailed composition of each PFT class as well as EF and D values for each considered tree species can be found in Simpson et al. (2012) (Sect 6.6., SI, S4.4).

For the European-domain runs used here, the EMEP model combines the PFT-specific EPs and max-LAI, with latitude-dependent growing season dates as described in Simpson et al. (2012). For this work we have made use of much finer grid-resolution (0.1°x0.1° latitude-longitude).

For non-forest vegetation types (e.g. grasslands, seminatural vegetation) or for forest areas not covered by the Köble and
Seufert (2001) maps (e.g. for eastern Russia), default emission factors taken from Simpson et al. (2012) were applied.

The crops category is the most difficult to deal with in terms of BVOC emissions, not least because the types of crops are not well known (and can change significantly over the years), and the growing seasons almost impossible to specify. Here we used a simple system which defines the phenology and emission factors of crops using EMEP model definitions. For this study, the





EMEP model's temperate crop (wheat, etc.), Mediterranean crop (maize, etc.) and root-crop (potato, etc.) were aggregated into one Crop PFT.

The isoprene emission potential data (EF$_{PFT}$) and the monthly changes in LAI per each of the six PFTs were provided for a European domain spanning [30.05° − 71.95°] in latitude and [-29.95° − 65.95°] in longitude with 0.1°x0.1° spatial resolution.

**2.5.2 Conversion of isoprene emission potential map for MEGAN**

In order to satisfy the MEGANv2.1 definition of standard conditions, the EMEP-based isoprene emission potential maps needed to be converted. As discussed earlier, there is unfortunately no precise way of such conversion. According to recommendations of Arneth et al. (2011) and Messina et al. (2016) the following equation was used for conversion between the EMEP and MEGANv2.1 system.


MEGAN-suitable isoprene emission potentials EP$_{MEGAN}$ (µg m$^{-2}$ h$^{-1}$) were calculated for each month as:

$$EP_{MEGAN}[month] = LAI_{std} \frac{\sum_i^{PFT} f_i \frac{LAI_i[month]}{LAI_{max}} EP_{PFT\ i}}{\sum_i^{PFT} f_i\ LAI_i[month]} \tag{4}$$


where $LAI_{std}$ is standard leaf area index in the MEGAN model equal to 5 m$^2$ m$^{-2}$, $i$ is an index through EMEP PFT categories (Table 3), $f$ is a fraction of a grid cell covered by specific PFT, $LAI$ and $LAI_{max}$ are monthly and maximal leaf area index of the PFT category (see Table 3), respectively, and $EP_{PFT}$ is EMEP-based isoprene emission potential for a specific PFT category.

Monthly isoprene emission potential maps for Europe were then embedded into the global domain of MEGAN gridded emission potential maps. These new global isoprene EPs were used in the calculation of the CAMS-GLOB-BIOv3.1 emission inventory.

**3 Results and discussion**

The following sections present examples of spatial and temporal distribution of emissions in CAMS-GLOB-BIO inventories
on global and regional scales (Sect. 3.1 and 3.2). Section 3.3 focuses in more detail on the impact of land cover change on isoprene emissions and Section 3.4 on update of the isoprene emission potential data in Europe showing differences between emissions calculated with the MEGAN-default and updated input EP maps. Section 3.5 presents comparison of CAMS-GLOB-BIO isoprene and monoterpene emissions to other available datasets.



### 3.1 Global distribution of BVOC emissions

The annual global totals averaged over the respective periods are listed in Table 4 for BVOC species available in CAMS-GLOB-BIOv1.2, v3.0 and v3.1 datasets. Though the absolute values differ between the datasets, the species responsible for the majority of the global BVOC total are common to all three inventories. The most abundant species is isoprene (64 %), followed by the sum of monoterpenes (13%), methanol (7%), acetone (4%), ethene (3.6%), sesquiterpenes (2.5%), propene (2%), acetaldehyde (1.4%) and ethanol (1.3%). The numbers in brackets represent the species contribution to the global BVOC

total when expressed as Tg(C) yr$^{-1}$, averaged over the three datasets. The rest of species contribute together with less than 10 Tg(C) yr$^{-1}$, i.e. less than 2%. Note that for monoterpenes we provide emissions of α-pinene, β-pinene and other monoterpenes. Other monoterpenes group is a sum of myrcene, sabinene, trans-β-ocimene, limonene and 2Δ-carene, following recommendations of Emmons et al. (2010).

The CAMS-GLOB-BIOv3.1 annual global total BVOC is about 60 Tg(C) yr$^{-1}$ higher than v1.2. This difference is mainly due to use of different meteorological inputs. While v1.2 was calculated with ERA-Interim reanalysis, v3.1 is based on ERA5. ERA-Interim data is available with 3 or 6 h time step and therefore to obtain hourly input fields for the MEGAN model, the data needed to be temporally interpolated. Such interpolation leads to underestimation of meteorological parameters, especially for air temperature and solar radiation, as the interpolated fields do not capture the noon peak hours at locations between the

model time steps. In the case of ERA5, the data are available with hourly timesteps and temporal interpolation is no longer needed. The annual mean ERA5 values of air temperature and solar radiation are therefore higher than ERA-Interim, especially in highly emitting regions of south America, central Africa, south-east Asia and Indonesia, which is reflected accordingly in higher modeled emissions. Similar effects of temporal resolution of the input climate data on isoprene emissions were discussed by e.g. Ashworth et al. (2010).


The largest difference in global emission total can be observed between the CAMS-GLOB-BIOv3.1 and v3.0 inventories. The v3.0 emission total is more than 160 Tg(C) yr$^{-1}$ lower than in v3.1, with most significant difference for isoprene estimates which are more then 140 Tg(isoprene) yr$^{-1}$ lower in v3.0 compared to v3.1. Both inventories are calculated with ERA5 meteorology, but they differ in setup of the input emission potential data and most importantly in the underlying land cover

description.

In the calculation of v3.0 we switched from using static CLM land cover maps to annually changing ESA-CCI land cover data in order to capture the effect of land cover change on emissions. To include the effect of changing land cover information in the model, input gridded emission potential maps (described in Sect 2.4.1) had to be replaced by calculation of emission

potentials from PFT distributions (described in Sect. 2.4.2). As discussed in Sect. 2.4.2, such a change in the model setup leads to ~10% decrease of isoprene emissions on global scale. The rest of the isoprene decrease can be explained by different land





cover distribution in CLM and ESA-CCI datasets. Sensitivity emission model runs using exactly same input data except for definition of land cover distribution resulted in isoprene annual global total of 427 Tg(isoprene) yr$^{-1}$ when using CLM and 316 Tg(isoprene) yr$^{-1}$ when using ESA-CCI, i.e. almost 30 % difference. As shown in Table 1, total vegetated area is more than 18

million km$^2$ (19%) smaller in ESA-CCI than in CLM maps. Significant differences between the two vegetation maps are visible in the tropical region which is a source of ~80% of isoprene global emission (Guenther et al., 2012; Sindelarova et al., 2014). The extent of broadleaf evergreen and deciduous tree cover in ESA-CCI is about 25% lower than in CLM.

The global spatial distribution of CAMS-GLOB-BIOv3.1 emissions for selected species is presented in Fig. 1. Regions of

highest emission are located in the tropical band and include Amazonia, central Africa south-east Asia, Indonesia and northern Australia.

**Table 4. List of modeled NMVOC species with annual global emission totals (Tg(species) yr$^{-1}$) in CAMS-GLOB-BIOv3.1, v3.0 and v1.2 inventory averaged over the dataset period. Each species / group is assigned**

**a molecular weight (right column) which was used to calculate total emissions in Tg(C) yr$^{-1}$.**

| species [Tg (species) yr$^{-1}$] | CAMS-GLOB-BIO.v3.1 2000 - 2019 | CAMS-GLOB-BIOv3.0 2000 - 2019 | CAMS-GLOB-BIOv1.2 2000 - 2017 | molecular weight [g mol$^{-1}$] |
|---|---|---|---|---|
| isoprene | 440.5 | 299.1 | 385.2 | 68 |
| α-pinene | 27.2 | 23.7 | 25.7 | 136 |
| β-pinene | 14.7 | 10.1 | 14.1 | 136 |
| other monoterpenes | 40.8 | 29.4 | 38.7 | 136 |
| methanol | 103.4 | 91.5 | 99.5 | 32 |
| acetone | 33.2 | 25.6 | 32.5 | 58 |
| acetaldehyde | 15.0 | 11.1 | 13.5 | 44 |
| formaldehyde | 3.7 | 2.9 | 3.4 | 30 |
| propane | 0.03 | 0.02 | 0.03 | 44 |
| propene | 13.3 | 10.9 | 13.0 | 40 |
| ethane | 0.28 | 0.23 | 0.27 | 30 |
| ethene | 23.5 | 19.2 | 21.9 | 28 |
| ethanol | 15.0 | 11.1 | 13.5 | 46 |
| sesquiterpenes | 16.6 | 11.9 | 14.9 | 204 |
| toluene | 1.2 | 1.0 | 1.1 | 92 |



| | | | | |
|---|---|---|---|---|
| MBO | 1.4 | 0.3 | 1.4 | 88 |
| formic acid | 2.8 | 2.2 | 2.5 | 46 |
| acetic acid | 2.8 | 2.2 | 2.5 | 60 |
| butanes and higher alkanes | 0.06 | 0.05 | 0.05 | 58 |
| butenes and higher alkenes | 2.7 | 2.2 | 2.6 | 56 |
| other aldehydes | 2.6 | 2.1 | 2.4 | 44 |
| hydrogen cyanide | 0.61 | 0.50 | 0.57 | 27 |
| hydrogen sulfide | 0.08 | 0.07 | 0.08 | 34 |
| other ketones | 0.6 | 0.5 | 0.6 | 72 |
| **total emissions** | | | | |
| *Tg (C) yr$^{-1}$* | **591** | **424** | **532** | |
| CO | 71.2 | 58.1 | 65.3 | 28 |











**Figure 1: Spatial distribution of emissions averaged over the 2000–2019 period for (a) isoprene, (b) sum of monoterpenes, (c) methanol, (d) sum of sesquiterpenes, (e) acetone and (f) ethene in the CAMS-GLOB-BIOv3.1 dataset.**

Additionally, significant BVOC emission sources are located in the south-eastern part of the US, especially during the summer months of the northern hemisphere. Substantial amounts of monoterpenes and methanol are further emitted from the northern temperate and boreal forests. BVOC emissions have strong seasonal variation following local meteorological conditions and vegetation cycle with highest emissions during daytime and in summer season and lowest emissions during nighttime and winter months.

Temporals variation of isoprene, the most abundant BVOC species, for the period of 2000-2019 from CAMS-GLOB-BIOv3.1 dataset are presented in Fig. 2. The plot shows global monthly totals and interannual variation of emissions as well as isoprene zonal means. The zonal means stress again the tropical and southern subtropical band as the most important source of global isoprene with additional source in northern temperate latitudes.

The interannual changes in emissions are driven partially by interannual changes in vegetation through changes in leaf area index, but to a greater extent by interannual changes in meteorology. There is a clear link between emissions and El Niño / La Niña phenomena. Annual global totals as well as zonal means in Fig. 2 show isoprene decrease in 2008 and 2011 when strong La Niña was identified and increase in 2002, 2015 and 2019 during El Niño episodes. Such a connection between BVOC emissions and El Niño phenomena was already noted in previous studies e.g. Naik et al. (2004), Müller et al. (2008) and in Wells et al. (2020).

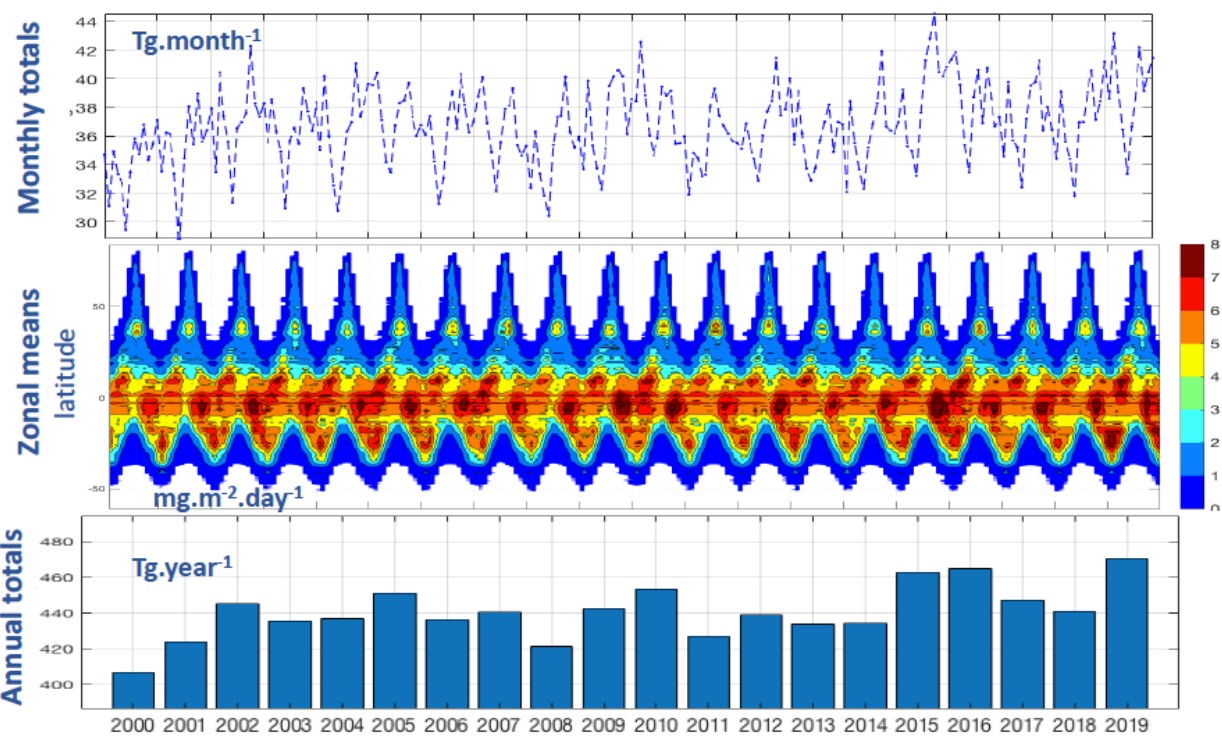

**Figure 2: Global monthly totals (top), zonal means (middle) and global annual totals (bottom) of the isoprene emissions for the period of 2000–2019 in the CAMS-GLOB-BIOv3.1 inventory.**

### 3.2 Regional distribution of emissions

The CAMS-GLOB-BIOv3.1 emissions of the main BVOC species for the year 2000 were further analyzed to show their regional contribution to global totals. We have used regions defined under the GlobEmission project

(https://www.globemission.eu/) which divide the globe to nine emitting areas. The spatial extent of the regions is given in Table 5 and shown in Fig. 3.

Table 5 presents the annual emission of isoprene, monoterpenes, methanol, acetone, sesquiterpenes and ethene from each of the regions together with their relative contribution to the global total. For all species (except for methanol), more than 70%

of emission originates in tropical regions of South America, East Africa and Southeast Asia and 10 – 18 % of emissions has its source in the northern latitudes (North America, Europe and Russia). Especially low, when compared to other species, is a production of isoprene in Europe and Russia, with less than 1% and 2 % of the global total, respectively. For methanol, the



tropics contribute with only 63% and almost 25% of methanol is produced in the northern latitudes, mainly in North America and Russia.


**Table 5. Regional annual emissions for CAMS-GLOB-BIOv3.1 isoprene, monoterpenes, methanol, acetone, sesquiterpenes and ethene expressed as *Tg(species) yr⁻¹* and as a percentage of the global total.**

| | Latitude extent | Longitude extent | Regional annual emissions [Tg(species) yr⁻¹ \| % of global total] | | | | | |
|---|---|---|---|---|---|---|---|---|
| *region* | *[latmin - latmax]* | *[lonmin - lonmax]* | isoprene | monoterpenes | methanol | acetone | sesquiterpenes | ethene |
| **North America** | 13°N - 75°N | 40°E - 170°E | 31 \| **8%** | 7 \| **9%** | 12 \| **12%** | 3 \| **10%** | 1.2 \| **8%** | 2.1 \| **9.5%** |
| **South America** | 60°S - 13°N | 35°E - 90°E | 133 \| **33%** | 27 \| **34%** | 23 \| **24%** | 10 \| **33%** | 6 \| **39%** | 6 \| **29%** |
| **Europe** | 36°N - 75°N | 50°W - 15°E | 3.6 \| **0.9%** | 2.8 \| **4%** | 5 \| **5%** | 1.1 \| **3%** | 0.2 \| **1.3%** | 0.7 \| **3%** |
| **N. Af. + Mid. East** | 15°N − 37°N | 65°W - 20°E | 6 \| **1.4%** | 0.5 \| **0.6%** | 1.4 \| **1.4%** | 0.3 \| **0.9%** | 0.1 \| **0.7%** | 0.3 \| **1.4%** |
| **East Africa** | 15°S - 15°N | 55°W - 20°E | 93 \| **23%** | 15 \| **19.5%** | 20 \| **21%** | 6.5 \| **21%** | 3 \| **21%** | 5 \| **24%** |
| **South Africa** | 15°S - 35°S | 55°W - 20°E | 13 \| **3%** | 2.5 \| **3%** | 4 \| **4%** | 1 \| **3%** | 0.3 \| **2%** | 1 \| **4.5%** |
| **Russia** | 37°N - 75°N | 50°W - 179°W | 7 \| **1.8%** | 4 \| **5%** | 7 \| **7%** | 1.5 \| **5%** | 0.4 \| **2.7%** | 1 \| **4.5%** |
| **Southeast Asia** | 10°S - 37°N | 65°W - 170°W | 74 \| **18%** | 15 \| **19%** | 18 \| **18.5%** | 6 \| **18%** | 3 \| **18.7%** | 4 \| **18%** |
| **Australia** | 10°S - 50°S | 65°W - 179°W | 46 \| **11%** | 4 \| **5.5%** | 7 \| **7%** | 2 \| **6%** | 1 \| **6.7%** | 1.4 \| **6%** |
| **Globe** | 89°S - 89°N | 179°E - 179°W | 407 | 78 | 97 | 32 | 15 | 22 |


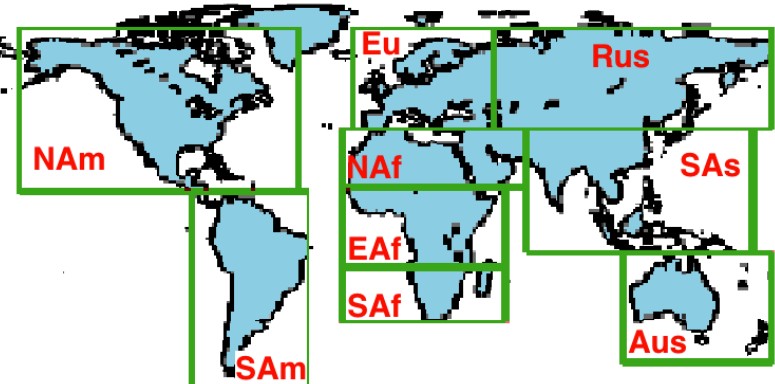

**Figure 3: Geographical extent of the GlobEmission regions. Adapted from GlobEmission (https://www.globemission.eu/).**




### 3.3 Impact of land cover change on isoprene emissions

The impact of changing land cover on emissions is captured in the CAMS-GLOB-BIOv3.0 dataset. To illustrate the effect of changing land cover on isoprene, the 20-year time series of isoprene annual totals was fitted with linear regression trend and compared to data from v3.1 for which a static CLM land cover map was used. When calculated with static vegetation map, isoprene emissions increase globally by 0.35 % yr$^{-1}$ due to temporal changes in meteorology. When annually changing ESA-CCI data are implemented, the trend decreases to 0.24 % yr$^{-1}$. Similar observation was made by Opacka et al. (2021), who

used a modified MODIS land cover data in the MEGAN-MOHYCAN emission model to study the impact of land cover change on isoprene emissions. They found a 0.04 to 0.33 % yr$^{-1}$ mitigating effect of land cover change on general positive trends of isoprene induced mainly by temperature and solar radiation.

Figure 4 presents a comparison of the v3.0 and v3.1 isoprene emission trends in selected regions. The regions' definition and

spatial extent is given in Table 5. Inclusion of land cover change through ESA-CCI data (v3.0) reduces isoprene trends in South America (esp. in the Amazon) or even causes a negative isoprene trend in Southeast Asia when compared to v3.1 based on static land cover. Such trend decline is caused mainly by retreat of tropical broadleaf forest (broadleaf evergreen and deciduous trees) in these locations. On the other hand, we observe an increase to isoprene trend in East Africa, North Africa + Middle East and Russia where the ESA-CCI data show an increase of broadleaf deciduous tree category (tropical and boreal)

in the course of the 20-year period. Moderate decrease to isoprene trend (relative difference of -20 to -30%) can be observed in North America, South Africa and Australia and the trend remains almost unchanged between the v3.0 and v3.1 data in Europe.







**Figure 4: Comparison of isoprene annual totals from CAMS-GLOB-BIOv3.0 and v3.1 in (a) South America, (b) Southeast Asia and Indonesia, (c) North America and (d) Europe with linear trend for each dataset (dashed line and trend value in [% yr⁻¹]).**

**3.4 Isoprene emission update in Europe**

Updated isoprene emission potential values in Europe, to more detail described in Sect. 2.5, were used to calculate isoprene emissions in CAMS-GLOB-BIOv3.1. Spatial distribution of annual mean isoprene emissions in Europe is presented in Fig. 5,

560



where CAMS-GLOB-BIOv3.1 emissions are compared with emissions obtained directly from the EMEP model (v4.33, Simpson et al., 2019) and with isoprene emissions calculated with MEGAN and v3.1 similar settings (i.e. meteorology, PFT distribution, LAI), but using the MEGAN-default emission potential maps instead of the updated EPs.

Figure 5 shows a good agreement in spatial distribution and amount of calculated emissions between the EMEP and v3.1 emissions which approves the approach of updated EP calculation and conversion from EMEP inputs to MEGAN format. It can also be seen that the spatial distribution of isoprene emissions changes when updated EPs are applied. Emissions calculated with MEGAN-default EPs are more uniformly distributed over the European domain, while v3.1 emissions are more localized, with isoprene hotspots in areas covered by highly-emitting tree species, e.g. in Portugal, Spain, southern France, and Balkan peninsula.

**Figure 5: Comparison of European annual mean isoprene emissions in 2016 from (a) the EMEP model and (b) CAMS-GLOB-BIOv3.1. Bottom panel (c) shows annual mean isoprene emissions calculated with default MEGAN EP maps.**



Use of updated instead of MEGAN-default EPs in Europe leads to 35% decrease of isoprene annual total from 10.03 Tg yr$^{-1}$ (v3.1 without EP update) to 6.55 Tg yr$^{-1}$ (v3.1). Isoprene monthly totals from these two datasets are compared to CAMS-GLOB-BIOv1.2 (annual total of 10.5 Tg yr$^{-1}$) and to isoprene estimated by EMEP (7.3 Tg yr$^{-1}$) in Fig 6. The plot shows a clear

decrease of emissions after use of the updated EPs and a good agreement between the CAMS-GLOB-BIOv3.1 and EMEP estimates. The results were extracted for the European domain with latitude from 30.5° to 71.75°, longitude from -29.75° to 65.75°.

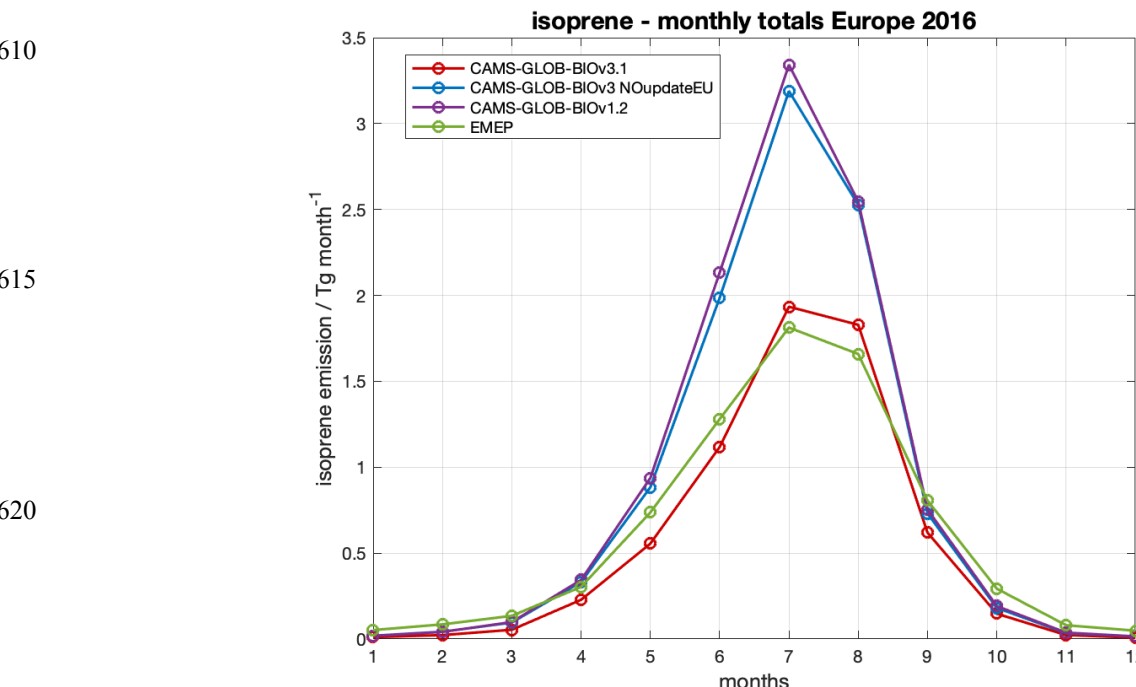

**Figure 6: Isoprene monthly totals in Europe in year 2016 from CAMS-GLOB-BIO datasets (v3.1, v3.1 without EP update and v1.2) and EMEP model.**

### 3.5 Comparison of CAMS-GLOB-BIO emissions with other inventories

Time series of CAMS-GLOB-BIO emissions of isoprene and monoterpenes were compared to other available data. We focus on isoprene and monoterpenes as these are the two most abundant BVOC species and the two species for which time series

from other sources are available. The rest of the species unfortunately suffers from lack of available and time-varying data.

Datasets gathered for this comparison are listed in Table 6. Each dataset is assigned with basic information such as model used for emission estimation and driving meteorology. Most of the inventories are so called 'bottom-up', i.e. modeled by an





emission model based on meteorology, emission factors and vegetation distribution. There are two 'top-down' datasets, IASB-
TD-OMI and IASB-TD-GOME2, which were calculated by an emission model and then constrained with satellite observations
       of formaldehyde (from OMI and GOME2) by applying an inversion technique in chemical transport model (IMAGESv2)
       (Stavrakou et al., 2014, 2015). Most of the inventories were calculated with 'MEGAN-like' emission model algorithm. Except
       for the GUESS dataset, which was estimated by a process-based model LPJ-GUESS (Smith et al., 2001; Sitch et al., 2003).
       The IASB datasets were obtained from website of the GlobEmission project (http://www.globemission.eu/). The rest of the
data were obtained from the ECCAD database (http://eccad.aeris-data.fr/).

**Table 6. List of datasets used for comparison of emissions.**

| *dataset* | *period* | *model* | *meteorology* | *inversion* | *reference* |
|---|---|---|---|---|---|
| **CAMS-GLOB-BIOv3.1** | 2000-2019 | MEGANv2.1 | ERA5 | - | This paper |
| **CAMS-GLOB-BIOv3.0** | 2000-2019 | MEGANv2.1 | ERA5 | - | This paper |
| **CAMS-GLOB-BIO.v1.2** | 2000-2017 | MEGANv2.1 | ERA-Interim | - | This paper |
| **MEGAN-MACC** | 1980-2017 | MEGANv2.1 | MERRA/MERRA2 | - | Sindelarova et al. (2014) |
| **IASB-TD-OMI** | 2005-2014 | MEGAN-MOHYCAN | ERA-Interim | OMI | Stavrakou et al. (2015) |
| **IASB-TD-GOME2** | 2007-2012 | MEGAN-MOHYCAN | ERA-Interim | GOME2 | Stavrakou et al. (2014) |
| **IASB-BU-OMI** | 2005-2014 | MEGAN-MOHYCAN | ERA-Interim | - | Stavrakou et al. (2015) |
| **GUESS** | 2000-2009 | LPJ-GUESS | CRU | - | Arneth et al. (2007a) |
| **MEGANv2** | 2003 | MEGANv2.0 | NCEP | - | Guenther et al. (2006) |


       Figures 7 and 8 show a comparison of isoprene and monoterpene annual totals within the 2000 – 2019 period, respectively. In
       both cases, the CAMS-GLOB-BIO emissions fall well within the range of other estimates. Though both plots show there is
       quite a large spread between the datasets with maximal differences up to factor of 2 to 3 (difference of 320 Tg yr$^{-1}$ for isoprene
       and 61 Tg yr$^{-1}$ for monoterpenes). There are various reasons for these discrepancies, with the most important being selection
of the emission model, driving meteorology and input vegetation data (land cover description and emission factors).

       In this data collection, the highest isoprene and monoterpene emissions are estimated by the MEGAN-MACC dataset. This
       dataset was calculated based on the MERRA/MERRA2 reanalyzes (Rienecker et al., 2011) and therefore differs in the use of
       meteorological inputs from most of the remaining datasets which used ERA meteorological fields (ERA-Interim and ERA5).
Comparison of 2 m temperature and PAR fields from MERRA and ERA datasets showed higher values of both meteorological
       parameters in the MERRA dataset mainly in the tropical regions of South America, central Africa and Australia, i.e. locations





of high BVOC emission potential. Higher temperature and PAR values in MERRA data then reflects in higher MEGAN-MACC estimates.

The key role of meteorology in BVOC estimation is supported also by a relatively good agreement of CAMS-GLOB-BIOv1.2 isoprene emissions with IASB datasets, esp. IASB-BU-OMI and IASB-TD-GOME2, which are both calculated based on ERA-Interim fields. Furthermore, the impact of meteorological inputs can be also observed in the difference between CAMS-GLOB-BIOv1.2 and v3.1 estimates, based on ERA-Interim and ERA5, respectively, which was already discussed in Sect. 3.1.

Inter-annual variability for most of the datasets is similar as they are mostly driven by the ERA meteorology. Again, except for MEGAN-MACC based on MERRA reanalysis for which the amplitude is higher. For all datasets there is a clear link between isoprene emissions and El Niño / La Niña phenomena. As also presented in Fig. 2, isoprene emissions decrease in 2008 and 2011 during strong La Niña and increase in 2002, 2015 and 2019 during El Niño episodes.

For monoterpenes the inter-annual variability is not as profound as for isoprene. Similar to isoprene, monoterpenes are strongly emitted in the tropical region. But have also significant sources in the temperate and boreal forests in the northern hemisphere. As a result, they are not as susceptible to atmospheric changes in the tropical band as isoprene and keep rather stable inter-annual profile.

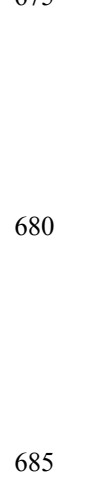

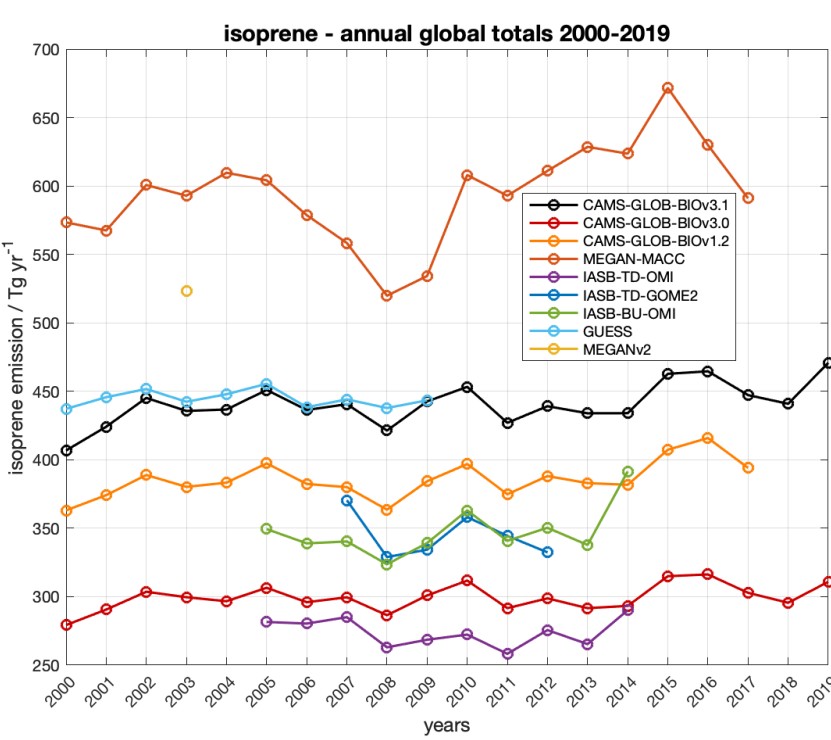




**Figure 7: Comparison of isoprene global annual totals from CAMS-GLOB-BIOv3.1 (in black), CAMS-GLOB-BIOv3.0 (red), CAMS-GLOB-BIOv1.2 (orange) and other available inventories within the 2000–2019 period.**


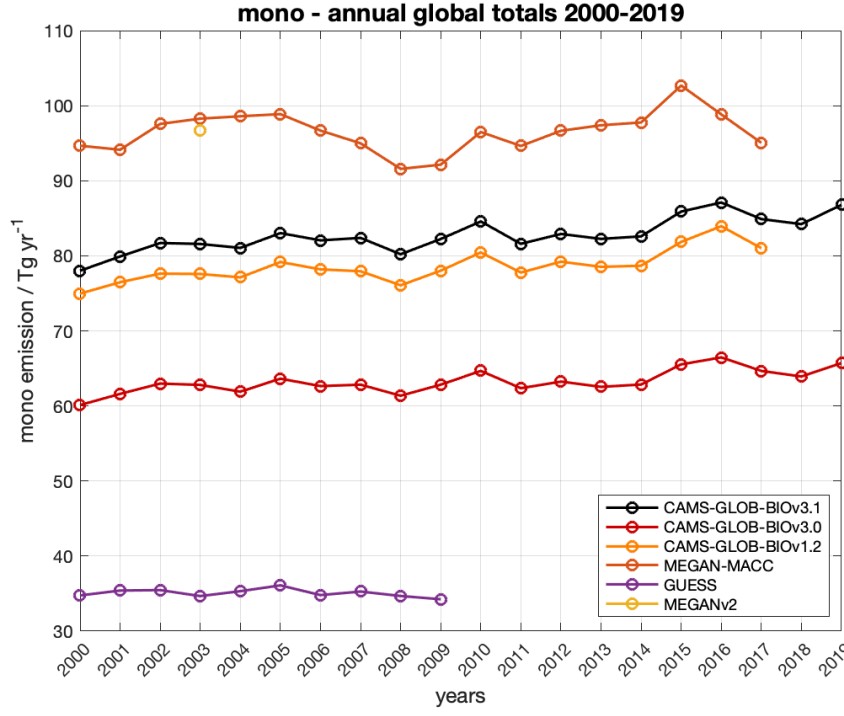




**Figure 8: Comparison of monoterpene global annual totals from CAMS-GLOB-BIOv3.1 (black), CAMS-GLOB-BIOv3.0 (red), CAMS-GLOB-BIOv1.2 (orange) and other available inventories within the 2000–2019 period.**

**4 Data availability**

Gridded maps with global emissions per species available as monthly means or monthly averaged daily profiles are provided as NetCDF (Network Common Data Format) files for the global domain at a resolution of 0.5° × 0.5° (CAMS-GLOB-BIOv1.2, https://doi.org/10.24380/t53a-qw03, Sindelarova et al., 2021a), and at a resolution of 0.25° × 0.25° (CAMS-GLOB-BIOv3.0, https://doi.org/10.24380/xs64-gj42, Sindelarova et al., 2021b; CAMS-GLOB-BIOv3.1, https://doi.org/10.24380/cv4p-5f79,

Sindelarova et al., 2021c) and can be accessed through the Emissions of atmospheric Compounds and Compilation of Ancillary Data (ECCAD) system with a login account (https://eccad.aeris-data.fr/, last access: June 2021). For review purposes, ECCAD



has set up an anonymous repository where subsets of the CAMS-GLOB-BIOv1.2, CAMS-GLOB-BIOv3.0 and CAMS-GLOB-BIOv3.1 data can be accessed directly (https://eccad.aeris-data.fr/essd-surf-emis-cams-bio/, last access: June 2021).

## 5 Conclusions

The presented paper describes three new global inventories of biogenic volatile organic compounds emitted from vegetation which are publicly available for use by the air quality and climate models. The datasets are called CAMS-GLOB-BIO v1.2, v3.0 and v3.1, and were calculated with the Model of Emissions of Gases and Aerosols from Nature (MEGANv2.1) driven by meteorological reanalyzes of the European Centre for Medium-Range Weather Forecasts (ECMWF).  Inventories include emissions of 25 BVOC species or chemical groups provided as monthly means and monthly averaged daily profiles spanning

through the period of 2000–2019. The CAMS-GLOB-BIO datasets were developed under the Copernicus Atmosphere Monitoring Service project (CAMS, Global and Regional Emissions) as part of the European Union's Copernicus Earth Observation Programme.

The dataset CAMS-GLOB-BIOv1.2 is based on ERA-Interim meteorological fields and is available with horizontal spatial

resolution of 0.5°x0.5°. Datasets CAMS-GLOB-BIOv3.1 and v3.0 were calculated with ERA5 meteorology and are provided with horizontal spatial resolution of 0.25°x0.25°.

The CAMS-GLOB-BIOv3.1 estimates global annual total BVOC emission of 591 Tg(C) yr$^{-1}$ with isoprene as the main contributing species (440.5 Tg(isoprene) yr$^{-1}$). Use of ERA5 meteorology in v3.1 leads to a slight increase of BVOC emissions

compared to v1.2 with global BVOC total of 532 Tg(C) yr$^{-1}$ (including 385.2 Tg(isoprene) yr$^{-1}$). The total emission in CAMS-GLOB-BIOv3.0 dataset is 424 Tg(C) yr$^{-1}$ with isoprene emissions of 299.1 Tg(isoprene) yr$^{-1}$. The difference between v3.1 and v3.0 estimates can mostly be attributed to use of an alternative land cover map for vegetation description.

The CAMS-GLOB-BIOv3.1 include isoprene estimates in Europe calculated with updated map of emission potential values

which are based on fine-scale land cover with detailed maps of tree species, and should therefore better represent the composition of European forests than the global EP maps of the MEGAN model. Use of updated isoprene EP maps led to a substantial decrease of European isoprene emission total by 35% and caused a change in spatial distribution of emissions. Isoprene emissions are concentrated in several emission hot spots in locations covered by highly-emitting tree species.

Both v3.1 and v1.2 estimates are based on a static land cover description obtained from the Community Land Model (CLM4). Since the world's vegetation is experiencing significant changes, such as deforestation in the tropical region, replacement of forests by agricultural land, afforestation efforts with fast-growing trees, we aimed to take this effect into account. Dataset CAMS-GLOB-BIOv3.0 considers changes in global land cover by using the ESA-CCI annual land cover maps for vegetation

description in the MEGAN model. In order to use a new land cover input in the model, the emission potentials had to be calculated from the PFT distribution instead of using the high-resolution emission potential maps. Such difference in input EP data and different input land cover map (ESA-CCI instead of CLM4) caused a decrease of ~30% for annual total isoprene and ~ 20% for monoterpenes when compared to CAMS-GLOB-BIOv3.1. The linear trend analysis of the 20-year time series of global isoprene emissions showed that inclusion of time-varying land cover data causes a decrease in general isoprene growing trend from 0.35 % yr$^{-1}$ to 0.24 % yr$^{-1}$. The trend slowdown is even more profound in the tropical regions of South America and Southeast Asia, where according to ESA-CCI data the retreat of tropical broadleaf forest can be observed. On the other hand, due to expansion of broadleaf deciduous trees the isoprene increasing trend is intensified in regions such as East and Central Africa or Russia.

Time series of CAMS-GLOB-BIO isoprene and monoterpene emissions were compared to other available data. The estimates fall well within the range of values from other studies. Though the comparison shows there is quite large uncertainty in emission estimates which on global scale can reach up to factor of 2 to 3, with even higher values on regional level. The different emission estimates in different versions of the CAMS-GLOB-BIO datasets suggest the uncertainty main driving factors, i.e. meteorological inputs, definition of input emission potentials and land cover distribution.

The presented CAMS-GLOB-BIO datasets provide a high-resolution data of global BVOC emissions for the period of 20 recent years based on up-to-date input data. The datasets are suitable for the purposes of air quality modelling, especially for models that do not include their own module for online BVOC emission estimation. Our general recommendation is to use a CAMS-GLOB-BIO dataset which is calculated with the same meteorology as drives the air quality model. If this does not apply, we recommend using the latest CAMS-GLOB-BIOv3.1 dataset. CAMS-GLOB-BIOv3.0 should be used for studies focusing on land cover change.

**Author contribution**

KS performed the emission model runs, created emission inventories and analyzed the data, JM contributed to preparation of model inputs and emission processing, DS provided data for the update of isoprene emission potentials in Europe, helped with its conversion to MEGAN model format and analysis of emissions, PH and JK participated on emission modelling, SD and CG performed the emission data formatting and upload to the emission database. KS wrote the manuscript with contributions from all co-authors.



**Acknowledgements**

The research leading to these results has received funding from the Copernicus Atmosphere Monitoring Service (CAMS), which is implemented by the European Centre for Medium-Range Weather Forecasts (ECMWF) on behalf of the European
Commission. The project was coordinated by the Centre National de la Recherche Scientifique (CNRS, Claire Granier) and by the TNO (Hugo Denier van der Gon).

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
