# Peer review of "High resolution biogenic global emission inventory for the time period 2000-2019 for air quality modelling"

_Earth System Science Data, 2021_

## Author Response (AR1)

**REVIEW 1**

We thank the anonymous reviewer for his/her kind review. We address the individual questions, comments and corrections (marked in blue) below.

This paper presents 3 new emission inventories for biogenic VOCs globally at a spatial resolution of X(**) and a time resolution of monthly for the years 2000 – 2019. The purpose of the new emission inventories is so that others can easily drive climate and air quality models without having to make the emission calculations themselves. The 3 inventories simulate biogenic emissions using:

- V2.1: ERA-Interim meteorology
- V3.1: The newer ERA-5 meteorology and updated isoprene emissions
- V3.0: ERA-5 and allowing land use changes over the period 2000 – 2019

V3.1 is the recommended dataset to use, unless the model being run uses ERA-Interim as input.

I do wonder whether monthly time resolution is enough for air quality models – I guess it depends on what the focus is of the study. If it's a day-to-day study of the impacts of biogenic emissions on city summertime smogs then monthly is probably too coarse a resolution. There can be wide fluctuations in emissions on day-to-day basis in summer. Some clarification is needed here.

The reviewer is indeed right that monthly temporal resolution is not enough to capture local and short-time fluctuations of the BVOC emissions. For such studies it is necessary to run the emission model with hourly time resolution. However, such simulations are costly to run on global level for a long period of several years, as well as represent a great demand for emission output data storage. The CAMS-GLOB-BIO emissions represent an approximation in this sense, however, they provide a longer-term picture of the BVOC emission evolution in time.

The paper is well written and is easy to follow. I only have some minor suggestions, and one possible error which needs checking.

**Actually, I'm not clear on the spatial resolution of the final products? The resolution of some of the input data is given, but unless I've missed it, I can't find a resolution in the body of the manuscript (Ok – the resolution appears in the data availability section at the end). The resolution should be included in the abstract at line 18 where 'high resolution' is mentioned.

The horizontal spatial resolution of each dataset is now mentioned in the abstract of the manuscript.

Line 75 + 77. First time we see these acronyms. MEGAN gets spelled out later in the next section, but might be worth doing here. Also LPJ-GUESS and JULES.

The text was edited and all acronyms are spelled out.

Line 155 'A' list of the MEGAN….

Corrected.

Line 170. Note that though 'the' corn….

Corrected.

Line 228. 'calculated from the EP maps'. Think that EP maps here actually should be EF maps? Check.

The EP maps are actually correct here. By EP maps we mean the high resolution input maps for α-pinene, where each grid cell is assigned with a value which represents a mean Emission Potential averaged over all vegetation types in that grid cell. In our notation, EF would be a value assigned to individual vegetation type.

Line 256. Just a side question – why is there only detailed land cover for Europe?

Compilation of detailed land cover data, preferably with information of individual tree species, and its assignment with emission factors for individual chem. species requires a lot of work by matching the land cover categories (tree species) with EF values available from the literature and/or EF measurement databases. The updated isoprene values for Europe used in this study are a result of work done for the EMEP model over many years. We continue working on the update of EF values also for other parts of the world. The limit is often availability of detailed enough land cover and availability of suitable EFs.

Line 315. I understand it is easy to aggregate all crops into one PFT, but then what emissions are 'crops' given? Eg I think corn is quite a high BVOC emitter compared to other crops, and could distort the average.

Indeed, the crops category is rather general and it contains species which are very high emitters as well as species with low or no emissions (NB this is true also for the forest categories). Though the CLM PFT categories distinguish between corn and other crops, the input land cover data that were available and that we used in the simulations, aggregate all crops into one category. Information on spatial distribution of corn only was not provided in the data. Therefore, the output emissions also provide values for the crop category as a whole.

Line 453 'temporal'

Corrected.

Line 456 'sources'

Corrected.

Figure 2. There looks to be an upward trend in the isoprene with time. It'd be interesting to comment or compare with the isoprene trend from the run where the land-use changes were taken into account (v3.0?). Do we think the upwards trend seen here is purely due to increasing temperatures?

The isoprene trend over time is discussed at the beginning of Section 3.3 where we also compare to the isoprene trends from the run with changing land cover.

"When calculated with static vegetation map, isoprene emissions increase globally by 0.35 % yr-1 due to temporal changes in meteorology. When annually changing ESA-CCI data are implemented, the trend decreases to 0.24 % yr-1. Similar observation was made by Opacka et al. (2021), who used a modified MODIS land cover data in the MEGAN-MOHYCAN emission model to study the impact of land cover change on isoprene emissions. They found a 0.04 to 0.33 % yr-1 mitigating effect of land cover change on general positive trends of isoprene induced mainly by temperature and solar radiation."

Opacka, B., Müller, J.-F., Stavrakou, T., Bauwens, M., Sindelarova, K., Markova, J., and Guenther, A. B.: Global and regional impacts of land cover changes on isoprene emissions derived from spaceborne data and the MEGAN model, Atmos. Chem. Phys., 21, 8413–8436, https://doi.org/10.5194/acp-21-8413-2021, 2021.

Table 5. Error in longitude extent which needs checking. At first I thought the east and west were the wrong way round. America is definitely west. Australia and south-east Asia are definitely east. SE Asia (India) starts around 67E and Australia 110E so the extents don't look correct either. Potentially values in this table aren't correct if these longitudes have been used in calculations.

The values of regional spatial extent were erroneously defined in the table, but were correctly used in the data analysis. The table was edited and corrected.

Line 512. Calculated with 'the' static….

Corrected.

Line 514. 'A' similar observation…

Corrected.

Line 606. Add 'E' and 'N' to the domain extents.

Corrected.

Figures 7 + 8. The red of the MEGAN-MACC and CAMS-GLOB-BIOv3 lines are very similar. I was initially confused by the statement that CAMS-GLOB-BIO fell within the range (I'd confused it with MEGAN-MACC). Tricky when there are lots of colors in play, but perhaps the CAMS-GLOB models could be shades of red/orange and MEGAN-MACC gets a different color? I also struggled to see the yellow MEGAN 2 line - but then realised it was just a circle - perhaps remove the yellow line from the legend? Ditto for monoterpenes plot.

The plots were edited so that MEGAN-MACC dataset is better distinguished from the CAMS-GLOB-BIO datasets and the MEGANv2 is now represented only with a dot in the legend.

**REVIEW 2**

We thank the anonymous reviewer for his/her kind review. Th specific question, comments and correctios (marked in blue) are addressed below.

**SPECIFIC COMMENTS**

Introduction, line 53: I agree that BVOC emissions from vegetation are driven by temperature, radiation, vegetation types and atmospheric composition ($CO_2$ concentrations for instance), but I don't see how atmospheric chemistry is driving those emissions, as stated in the text. Could the authors clarify this point ?

By atmospheric chemistry driver at this point we meant processes connected to the formation of tropospheric ozone. When emission of VOCs, via formation of peroxy radicals, impacts concentration of low level ozone, which on the other hand impacts VOC emissions by causing oxidative stress on the plants (e.g. Pinto et al., 2010). High $O_3$ concentrations may cause damage to plants stomata and decrease photosynthetic rate, therefore decrease VOC emissions. $O_3$ also activates plants production of the reactive oxygen species, which in turn stimulate various defense mechanism depending on the plant species (e.g. Li et al., 2017).

We realize that this effect can be more simply described as impact of "atmospheric composition" which includes also $O_3$. Therefore "atmospheric chemistry" was omitted from the text.

Pinto, D.M., Blande, J.D., Souza, S.R. et al. Plant Volatile Organic Compounds (VOCs) in Ozone ($O_3$) Polluted Atmospheres: The Ecological Effects. J Chem Ecol **36,** 22–34, https://doi.org/10.1007/s10886-009-9732-3, 2010.

Li, S., Harley, P., Niinemets, Ü.: Ozone-induced foliar damage and release of stress volatiles is highly dependent on stomatal openness and priming by low-level ozone exposure in Phaseolus vulgaris, Plant, Cell & Environment, Vol. 40, Issue 9, https://doi.org/10.1111/pce.13003, 2017.

Methodology/Emission model, line 115: it is stated that MEGAN, emission model used for this study, calculates BVOC emissions from vegetation and soils. Later in the manuscript, biogenic emissions are presented and described essentially as vegetation emissions. Is the soil contribution actually taken into account in the estimates given and activated in the model for this work, and could the authors precise the order of magnitude of these emissions, compared to vegetation ones ? If considered, how are soil emissions calculated in the model? Do they follow the same dependency than vegetation emissions?

No, soil emissions were not calculated in this study and are not part of the CAMS-GLOB-BIO dataset. At this point of the manuscript, we mention soil emissions as there is an option in the MEGAN model to calculate NOx emissions from soils and some of the cited studies present estimates of these emissions. But calculation of soil NOx emission was not activated in our study.

Methodology, lines 225-227: The impact of using EP calculated from PFT coverage is specified for isoprene (-10%) and other compounds. Where does this estimate come from? Is it a general understanding (and if so, relevant papers should be cited) or did the authors run specific simulations to provide this estimate (if so, this should be clarified).

To evaluate the difference between the emissions calculated based on EP detailed maps and on EP calculated from PFT coverage, we performed specific emission model runs. To clarify this, the following two sentences were added to the text of the manuscript on line 225.

"We performed specific emission model runs to evaluate the difference in resulting emissions when emissions are calculated from EP detailed maps and from EP calculated based on the PFT coverage. All the other input parameters were kept the same. Use of EP calculated from the PFT coverage leads to ~10% decrease of isoprene emission total on global scale…"

Biogenic VOC emissions are driven in particular by temperature and radiation. Therefore, they are strongly variable over the course of a day, and can vary significantly from one day to another. For climate or air quality modeling purposes, considering as forcings monthly means or monthly averages daily profiles (even if subsequently interpolated hourly) instead of hourly emissions could impact the results of the investigations. Can the authors give an estimate of this impact, considering ozone for instance?

Indeed, temporal resolution of the input meteorological data has an impact on the resulting emissions. Ashworth et al. (2010) evaluated differences in the calculated emissions between emission model runs driven by hourly, daily average and monthly average data. They found that using monthly mean meteorology instead of hourly leads to ~ 7 % reduction of the global annual isoprene emissions, with local reductions up to 55%.

Estimating the impact of such emission difference on the tropospheric ozone concentrations requires inclusion of the atm. chem. model, as the impact will depend on other atm. conditions such as NOx concentration and whether the atmosphere is in the VOC or NOx limited state. However, such evaluation is out of the scope of the presented paper.

Ashworth, K., Wild, O., and Hewitt, C. N.: Sensitivity of isoprene emissions estimated using MEGAN to the time resolution of input climate data, Atmos. Chem. Phys., 10, 1193–1201, https://doi.org/10.5194/acp-10-1193-2010, 2010.

A great effort has been made in improving and updating emission factors for the different PFT classes considered in the model. Could the authors specify if/how biofuel crops are considered? As usually high BVOC emitters, especially regarding isoprene, these vegetation species could strongly impact global and regional emissions, but are not always considered, and easy to consider, in emission models.

No, biofuels are not considered. The main basis for European BVOC emissions in Europe in the EMEP system is a map of forest species generated by Köble and Seufert (2001), which was based upon surveys from the 1990s. As in Simpson et al. (1999), isoprene emissions from crops were neglected, based upon some low emission rates reported in the literature, but also as emissions from forests were believed to dominate.

Inclusion of biofuels would of course be a good improvement in principle. Although emission factors for some species can be specified, the major problem would be to specify the spatial distribution, phenology and agricultural practices related to such vegetation. This is likely a major effort, and should be combined with a general update in the underlying forest-species maps where possible.

Köble, R. and Seufert, G.: Novel Maps for Forest Tree Species in Europe., in: A Changing Atmosphere, 8th European Symposium on the Physico-Chemical Behaviour of Atmospheric Pollutants, Torino, Italy, 17-20 Sept., 2001.

Simpson, D., Winiwarter, W., Borjesson, G. et al., Inventorying emissions from Nature in Europe, J. Geophys. Res., 104, 1080 8113-8152, 1999.

**TECHNICAL CORRECTIONS**

Line 44: for a better reading replace at the beginning of the paragraph "Their oxidation" by "BVOC oxidation". Same line, replace "an important role in formation of low-level ozone" by "an important role in the formation of low-level ozone".

Corrected.

Line 71: change the second sentence to "The models differ in the approach used to estimate BVOC, in the level of complexity in processes considered and in factors affecting the emission".

Corrected.

Line 96: replace "calculated with modified version" by "calculated with a modified version".

Corrected.

Line 178: replace "Yuan et al." by "Yuan et al. (2011)".

Corrected.

Line 254-255: remove "to" in "maps are well suited to for the tropical region".

Corrected.

Line 311: replace "The crops category" by "The crop category".

Corrected.

The different inventories v3.0, v3.1 and v1.2 are characterized each by specific conditions for spatial resolution, time-period, land-cover map and emission factors/potentials, which are given in different locations in the text but are not easy to put together. The simulation description would really gain in clarity by adding a table presenting the different conditions for the 3 inventories.

Table summarizing and describing each dataset was added in the text for the reader to better see the characteristics of each dataset.

Table 4, caption: please specify what NMVOC means (not used anywhere else in the manuscript).

The acronym NMVOC (non-methane volatile organic compounds) was replaced by BVOC (biogenic volatile organic compounds) to harmonize with the rest of the text.

Line453: replace "Temporals variation" by "Temporal variations".

Corrected.

Line 566: replace "to more detail described in Sect. 2.5" by "described in more detail in Sect. 2.5".

Corrected.